# Adherence to Mediterranean Diet, Dietary Salt Intake, and Susceptibility to Nephrolithiasis: A Case–Control Study

**DOI:** 10.3390/nu16060783

**Published:** 2024-03-09

**Authors:** Veronica Abate, Anita Vergatti, Paola Iaccarino Idelson, Costantino Recano, Marzia Brancaccio, Domenico Prezioso, Riccardo Muscariello, Vincenzo Nuzzo, Gianpaolo De Filippo, Pasquale Strazzullo, Raffaella Faraonio, Ferruccio Galletti, Domenico Rendina, Lanfranco D’Elia

**Affiliations:** 1Department of Clinical Medicine and Surgery, Federico II University, 80131 Naples, Italy; veronica.abate@unina.it (V.A.); anita.vergatti@unina.it (A.V.); paola.iaccarinoidelson@gmail.com (P.I.I.); recano.costantino@hotmail.it (C.R.); mars.brancaccio@gmail.com (M.B.); galletti@unina.it (F.G.); lanfranco.delia@unina.it (L.D.); 2Department of Neuroscience, Reproductive Sciences and Dentistry, Federico II University, 80131 Naples, Italy; domenico.prezioso@unina.it; 3Endocrinology and Nutrition Department, Ospedale del Mare, 80147 Naples, Italy; drmuscariello@gmail.com (R.M.); vincenzo.nuzzo@aslnapoli1centro.it (V.N.); 4Assistance Publique-Hôpitaux de Paris, Hôpital Robert Debré, Service d’Endocrinologie et Diabétologie, 75019 Paris, France; gianpaolo.defilippo@aphp.fr; 5Former Professor of Internal Medicine, Federico II University, 80131 Naples, Italy; strazul@unina.it; 6Department of Molecular Medicine and Medical Biotechnology, Federico II University, 80131 Naples, Italy; raffaella.faraonio@unina.it

**Keywords:** dietary habits, MEDILITE, MINISAL, nephrolithiasis

## Abstract

Unhealthy dietary habits play a key role in the pathogenesis of nephrolithiasis (NL). The aims of this case–control study were to evaluate (i) the adherence to the Mediterranean Diet (MD) and the dietary salt intake in stone-forming patients (SF), (ii) the relationship occurring between MD adherence, salt intake and NL-related metabolic risk factors in SF, and (iii) the impact of combined high MD adherence and low salt intake on NL susceptibility. From 1 January 2018 to 31 December 2019, we recruited all SF consecutively referred to the Extracorporeal Shock Wave Lithotripsy (ESWL) center of Federico II University, and at least two control subjects without a personal history of NL, age-, sex-, and body mass index-matched to SF (NSF). All study participants were interviewed using the validated MEDI-LITE and MINISAL questionnaires. In an SF subgroup, the NL-related metabolic risk factors were also evaluated. SF showed a lower MD adherence and a higher salt intake compared with NSF. The NL susceptibility decreased by 36% [OR: 0.64 (0.59–0.70); *p* < 0.01] for each point of increase in MEDI-LITE score, while it increased by 13% [OR: 1.13 (1.03–1.25); *p* = 0.01] for each point of increase in MINISAL score. The SF prevalence was higher among subjects showing combined low MD adherence and high salt intake. In SF, the MEDI-LITE score directly correlated with 24 h-citraturia, whereas the MINISAL score directly correlated with urinary sodium and uric acid excretion. In conclusion, high MD adherence and low salt intake are associated with a reduced NL susceptibility, both separately and in combination.

## 1. Introduction

Nephrolithiasis (NL) affects between ten and fifteen percent of the world’s population and it is a significant worldwide source of morbidity, in particular for hydronephrosis and chronic kidney disease (CKD) [1]. Several epidemiological studies have demonstrated a rising incidence of NL over the last few decades and a high rate of recurrence after an initial stone event [2,3,4]. Indeed, the reported recurrence rate ranges from 6.1% to 66.9% and more than ten percent of patients could experience more than one relapse [5]. For these reasons, primary and secondary prevention plans for NL are part of public health programs [6,7,8,9,10,11]. 

Unhealthy dietary habits are acknowledged to be an important risk factor for NL and for its recurrence [10,11,12]. Indeed, many studies have found an association between the consumption of certain foods and the promotion or inhibition of NL [13,14,15,16]. However, dietary patterns that also consider the frequency and combination of certain foods may provide more exhaustive information on the impact of nutritional prevention on NL, rather than single food consumption [17]. In this regard, a few recent prospective studies have detected a protective role for NL of both the Mediterranean diet (MD) and Dietary Approaches to Stop Hypertension (DASH) diet, probably because of the large prevalence in both dietary models of plant foods compared to meat [16,17,18,19,20,21]. 

Furthermore, the diet influences the urinary acidic load (pH), which is a well-known risk factor for NL [22]. Indeed, a more alkaline diet, so one that is rich in fruit and vegetable and poor in meat and sweets intake, is related to more alkaline urine [23]. At the same time, a higher urinary pH is related to the crystallization of stones containing calcium and phosphate. The opposite is that a lower urinary pH promotes uric acid or cystine stones [24]. 

On the other hand, the beneficial effect of lower sodium intake on NL risk has been known for a very long time [25]. Indeed, the urinary excretion of calcium, which is the main stone compound, is influenced by the concurrent degree of sodium excretion in a ratio of one to one hundred, respectively [26]. 

In principle, patients who have experienced a single NL episode are expected to be particularly focused on dietary preventive measures, in order to reduce the risk of kidney stone recurrence. Nevertheless, data are limited about the degree of adherence to beneficial plant-based dietary models, such as MD, and about the tendency to reduce sodium intake by NL patients. Also, some methods are useful to evaluate the diet composition, for example non-invasive laboratory tests (i.e., evaluating urinary pH) [23], as well as validated questionnaires, exploring dietary habits. Based on these considerations, we performed a case–control study, aiming to evaluate (i) MD adherence and the dietary salt intake in stone-forming patients (SF) and in non-stone forming controls (NSF); (ii) the relationship occurring between MD adherence, salt intake, and NL-related metabolic risk factors; and (iii) the impact of combined high MD adherence and low salt intake on NL susceptibility. We enrolled SF treated with extracorporeal shock wave lithotripsy (ESWL), which is frequently used as a cost-effective treatment for NL [25,27,28,29,30], and NSF living in a geographical area that features a high incidence of NL: the Campania region in Southern Italy [25]. 

## 2. Materials and Methods

### 2.1. Study Populations

We enrolled all SF referred to the ESWL Center of Federico II University from 1 January 2018 to 12 December 2019. In the same lapse of time, at least two age-, sex-, and body mass index (BMI)-matched NSF for each SF were enrolled as control group. NSF were enrolled among employees of Federico II University Hospital and of the Ospedale del Mare, both located in Naples, and among the chaperones of patients referred to both Hospitals, Federico II University and Ospedale del Mare. All the study participants were interviewed about their habitual diet, according to the previously validated MEDI-LITE and MINISAL questionnaires. The MED-LITE explores the degree of adhesion to a Mediterranean diet [31]; on the other hand, MINISAL explores the dietary salt habits [32]. Inclusion criteria for the SF group were age > 18 years and the occurrence of calcium kidney stones eligible for ESWL treatment according to the American Urology Association guidelines [33,34]. Exclusion criteria for both SF and NSF groups were age < 18 years, primary or secondary hyper/hypoparathyroidism [34,35,36,37], hyper or hypothyroidism [38], gout diathesis [39], chronic diarrheal conditions, body mass index ≥ 35 Kg/m^2^ or ≤17.5 Kg/m^2^, ongoing cancer disease, protein malnutrition, type 1 and type 2 diabetes mellitus, incomplete data collection, and no consent to the nutritional interview. Informed consent was obtained from all participants included in the present study. The study was approved by the local Ethical Committee (protocol number 115/20). 

### 2.2. Data Collection

For both SF and NSF, the following data were collected in a standardized and pre-piloted schedule: age; sex; weight; height; degree of education; degree of adhesion to MD; estimation of the daily salt intake; group for combined MD and salt intake estimation; and urinary metabolic NL risk factors when available. For education, a dichotomous variable was used indicating whether or not the subject had obtained an upper secondary school diploma. 

### 2.3. MD Adherence

The degree of adhesion to MD was assessed using the validated nine-item questionnaire MEDI-LITE [31]. The questionnaire explores the habitual consumption of nine food groups or beverages considered crucial to assess the degree of adhesion to MD: (1) fruit; (2) vegetables; (3) cereal grains; (4) legumes; (5) fish and fish products; (6) meat and meat products; (7) dairy products; (8) alcohol intake; and (9) olive oil. For fruits, vegetables, cereals, legumes, and fish groups, 2 points are assigned for the category with the highest, 1 point for the category with the medium, and 0 points for the category with the lowest consumption. As for the olive oil group, 2 points are assigned for its regular use, 1 point for frequent use, and 0 points for only occasional use. For meat and dairy products, 2 points are assigned for the category with the lowest, 1 point for the category with the medium, and 0 points for the category with the highest consumption. Finally, 2 points are assigned for medium alcohol consumption (i.e., from 1 to 2 alcohol units per day), 1 point for lowest (1 alcohol unit per day), and 0 points for the category with the highest (more than 2 alcohol units per day). The final MEDI-LITE score is the sum of all the individual food scores and can range from 0 (minimum adhesion to MD) to 18 (maximum adhesion to MD) [31]. Points assigned to each question and the final score were included in the data collection. 

### 2.4. Salt Habits

To obtain an estimation of the daily salt intake, the MINISAL questionnaire was administered to all the study participants [32]. The questionnaire includes the following questions: (1) How often do you use salt at the table? (2) How much bread do you eat in one day? (bread representing the major source of salt in the average Italian diet [40]) (3) How many times a week do you eat cheese and/or cold cuts (these being other important sources of salt intake)? (4) Do you ever get thirsty, especially after a meal? (5) When you eat out of home, food seems usually salty, normal, or bland? Each question had 3 possible answers, with a score increasing from 1 to 3 for questions from 1 to 4. For the last question, 3 points were assigned in case of a bland perception, 2 for normal, and 1 for salty. The total MINISAL score is given by the sum of the individual scores and can range from 5 (lowest estimated salt consumption) to 15 (very high estimated salt consumption). Points assigned to each question and the final score were included in the data collection.

### 2.5. Combined Impact of MD Adhesion and Salt Intake on NL Susceptibility

To evaluate the combined impact of MD adherence and salt intake on NL susceptibility, the whole study population was dichotomized according to median values of MED-LITE and MINISAL scores. Thus, four groups featuring different dietary habits were obtained. The groups were organized and named as follows: (i) high MD adhesion and low salt intake (group A); (ii) high MD adherence and high salt intake (group B); (iii) low MD adherence and low salt intake (group C); (iv) low MD adherence and high salt intake (group D). The group for each patient was included in the data collection.

### 2.6. NL-Related Metabolic Risk Factors

The NL metabolic risk factors were estimated in the first forty-eight SF patients sequentially enrolled. To achieve this purpose, 24 h urinary samples were obtained from the SF group. For both collections, patients were advised to maintain their habitual diet and fluid load. Twenty-four hour urine samples were collected and analyzed for calcium (uCa), magnesium (uMg), phosphate (uPO4), sodium (uNa), potassium (uK), citrate (uCit), oxalate (uOx), and urate (uUr) concentrations. Urinary volume (uV) from 24 h collection was also determined. 

### 2.7. Statistical Analysis

The statistical analysis was carried out using the Statistical Package for Social Science (SPSS) for Windows, version 28 (SPSS Inc., Chicago, IL, USA). To assess possible differences in the characteristics of SF and NSF groups, we used Student’s *t*-test with Bonferroni correction for continuous variables and the chi-squared test for categorical variables. The Mann–Whitney test was used to evaluate the difference in MEDI-LITE and MINISAL scores between the study groups. The results were expressed as absolute values, percentage, or mean ± standard deviation, as appropriate, unless otherwise indicated. A multivariable linear regression analysis was carried out to determine the independent association between (log-transformed) urinary NL risk factors and MEDI-LITE and MINISAL scores, adjusting for the main potential confounders. Two-sided *p* values lower than 0.05 were considered statistically significant. 

## 3. Results

In the entire study period, we enrolled 255 SF and 535 NSF. The anthropometric characteristics of both study groups are shown in Table 1. Among SF, there was a higher prevalence of subjects who had obtained an upper secondary school diploma. According to inclusion and exclusion criteria, no statistically significant difference was observed for age, gender, or BMI. 

### 3.1. MEDI-LITE Score

The MEDI-LITE score distribution in SF and NSF is shown in Figure 1. The answers to each item of the MEDI-LITE questionnaire items provided by the participants in the study cohorts are shown in Table 2. 

The mean MEDI-LITE score was significantly lower in SF compared to NSF (9.71 ± 2.11 vs. 11.55 ± 1.97; *p* < 0.01). Based on the analysis of each single food group, SF ate the recommended portions of fruit (*p* = 0.01), vegetables (*p* < 0.01), cereals (*p* < 0.01), legumes (*p* < 0.01), and fish (*p* < 0.01) less frequently, and used olive oil less regularly (*p* = 0.01) compared to NSF. On the other hand, SF ate dairy products (*p* = 0.01) and drank alcoholic beverages (*p* = 0.01) more frequently than NSF. A multivariate model indicated that, for each point of increase in the MEDI-LITE score, the probability of being an SF patient decreased by 36% [OR: 0.64 (0.59–0.70); *p* < 0.01]. This finding was confirmed after adjustment for BMI, age, and degree of education [37%, OR: 0.63 (0.58–0.69); *p* < 0.01]. 

### 3.2. MINISAL Score

The distribution of the MINISAL score in SF and NSF is shown in Figure 2. The answers to each item of the MINISAL questionnaire obtained in the study groups are shown in Table 3.

The mean MINISAL score was significantly higher in SF compared to NSF (9.01 ± 1.23 vs. 8.65 ± 1.57; *p* = 0.04). SF used salt more frequently at the table (*p* < 0.01), consumed more bread (*p* = 0.01) and cheese and/or cold cuts (*p* < 0.01), and were thirstier after a meal (*p* < 0.01) compared to NSF. Also, SF perceived a meal eaten out of the home as saltier compared to NSF (*p* < 0.01). A multivariate model indicated that, for each point of increase in the MINISAL score, the probability of being an SF patient increased by 13% [OR: 1.13 (1.03–1.25); *p* = 0.01]. This finding also remained significant after correction for BMI, age, and degree of education [OR 1.13 (1.02–1.25), *p* = 0.02].

### 3.3. Combined MD Adhesion and Salt Consumption

The median values of MEDI-LITE and MINISAL scores in the whole study population were 11 and 9, respectively. After the dichotomization of SF and NSF according to the median values of MEDI-LITE and MINISAL scores, we obtained the groups reported in Table 4. Among study participants with a MEDI-LITE score > 11 and MINISAL score < 9, Group A in Table 2, we found a higher prevalence of NSF and a lower prevalence of SF. On the other hand, among study participants with a MEDI-LITE score < 11 and MINISAL score > 9, Group D in Table 2, we found a lower prevalence of NSF and a higher prevalence of SF. Study participants belonging to B and C groups showed an intermediate prevalence of NSF and SF.

### 3.4. Correlation between MEDI-LITE Score, MINISAL Score, and NL Metabolic Risk Factors

The NL-related metabolic risk factors, were evaluated in 48 SF patients (22 men and 26 women; mean age: 51.9 ± 11.2 years; BMI: 26.9 ± 5.12 Kg/m^2^). The biochemical results from the 24 h urinary samples were uCa = 4.78 ± 2.55 mmol/24 h, uMg = 3.32 ± 1.51 mmol/24 h, uPO4 = 21.9 ± 7.4 mmol/24 h, uNa = 135 ± 53 mmol/24 h, uK = 48 ± 36 mmol/24 h, uCit = 2.99 ± 2.13 mmol/24 h, uOx 246 ± 161 mmol/24 h, and uUr = 2.72 ± 1.04 mmol/24 h. The 24 h urinary volume was 1.58 ± 0.53 L/24 h. Evaluating the relationship occurring between MEDI-LITE and MINISAL scores and NL-related metabolic risk factors, we detected a positive relationship between MEDI-LITE and uCit (r = 0.356, *p* = 0.013), and between MINISAL and uNa (r = 0.322, *p* = 0.026) and uUr (r = 0.322; *p* = 0.035). The results were also confirmed after adjustment for BMI, age, and gender.

## 4. Discussion

Unhealthy dietary habits play a key role in the pathogenesis of kidney stones and their risk of recurrence. The present study of the dietary habits of SF and NSF living in the same geographic area demonstrates that SF patients have a lower degree of adhesion to MD together with higher dietary salt consumption compared to NSF, age- and sex-matched. These results suggest an additive effect of both these unhealthy dietary habits on the susceptibility to NL. The study results also indicate that, in SF, the degree of adhesion to MD directly correlates with uCit, whereas the salt consumption directly correlates with uNa and uUA. Citrate is a strong inhibitor of stone formation by decreasing the supersaturation of calcium salt [41]. Previous studies indicate that citrus and non-citrus fruits are natural sources of dietary citrate, and that their elevated dietary intake significantly increases uCit and uK, thus lowering the NL risk [17,42]. On the other hand, higher salt consumption has been associated with higher uUA, which is involved in the nucleation and crystallization of calcium and/or urate kidney stones [43,44,45]. 

To date, the dietary habits in SF patients have been investigated separately for MD adherence and salt dietary intake. Among these investigations, Rodriguez et al. documented the reduction in NL risk in participants with the highest score of MD adherence, and so aligned with our results [25]. On the other hand, the association between salt intake and NL risk was a long-known finding [46]. Indeed, the DASH diet is strongly recommended in this setting [20]. The novelty of the present study is represented by the combined evaluation of both dietary habits at the same time and in the same study population, proving that both are needed for such a condition and one must not exclude the other. In addition, this study fits perfectly in the Italian background, both because of the high incidence of kidney stones [25] and because of the high salt consumption [47], even with the strongest MD adherence [48].

In the present study, among SF, we found an equal prevalence of men and women, at variance with previous evidence of a higher incidence of NL [49]. It is possible that the higher-than-expected prevalence of women among SF patients in our study reflects a global trend to a progressive increase in the number of women affected by NL observed in recent decades [50,51,52]. The increased prevalence of NL occurs in parallel with the increased prevalence of metabolic syndrome and its constitutive elements, in particular obesity [53,54]. Both metabolic syndrome and obesity are recognized risk factors for NL [55]. A similar pattern was also found in other conditions affecting the mineral metabolism, such as osteoporosis, which can be considered the other side of the same process, as well as a risk factor for NL [56]. The association with metabolic syndrome was found in this case in women but not in men [57,58], so the current degree of adhesion to the MD pattern observed, in Western countries, may be one of the reasons for the growing incidence of metabolic syndrome, osteoporosis, and NL [59]. 

Although a not-univocal relationship has been revealed on different occasions in the literature, our data showed a synergic favorable effect of the combination of MD and low salt intake. Indeed, whilst plant-based diets are low or very low in salt content, discretionary salt seasoning and many processed foods with high salt content are common [60]. 

A strength of the present study is that the degree of adhesion to MD and the estimation of salt consumption were performed using two validated questionnaires. The simultaneous evaluation of both has rarely been performed and, to date, never in SF patients [60]. A confirmation of the robustness of the study results comes from the detection of a direct relationship observed between the MINISAL score and 24 h urinary sodium excretion, considered as a marker of dietary salt intake [60]. The cross-sectional design offers the chance to evaluate two large and well-comparable populations, enrolled during the same lapse of time and in the same geographical area. On the other hand, it impairs the establishment of a cause–effect relationship between adhesion to both healthy dietary habits (MD and low salt intake) and susceptibility to NL. Our findings, however, encourage the future performance of a randomized controlled trial (RTC) to validate the study hypothesis and to establish a possible causal relationship. With this aim, supplementing the data obtained from the questionnaires with biochemical data, such as urinary pH, could strengthen the conclusions. The failure to assess the NL metabolic risk factors in the entire population is another study limitation that can be justified by its design.

## 5. Practical Application

The results from this study indicate the need for intensive action aimed to increase the patients’ awareness about the dietary intervention measures known to be effective in the prevention of NL [61]. In this regard, a multidisciplinary management of SF patients is desirable. It should include the general practitioner and the consultant urologist to treat the kidney stone occurrence (for example, through ESWL and pain medication); a mineral metabolism expert to provide a comprehensive view of the condition and the related risk factors; and the nutritional specialist to educate patients on these two effective behaviors to prevent possible future events. Nutritional recommendations should focus on both a Mediterranean pattern diet and salt restriction. 

## 6. Conclusions

In conclusion, the study results indicate that SF patients have a lower adhesion to MD and a higher dietary salt consumption compared to controls. These results, which should be confirmed and supported by a randomized controlled trial, suggest the need for more intensive action from the general practitioner in the prevention of NL, based on nutritional recommendations by a specialist, including both a better adhesion to a Mediterranean dietary pattern and a substantial reduction in dietary salt intake [61]. 

## Figures and Tables

**Figure 1 nutrients-16-00783-f001:**
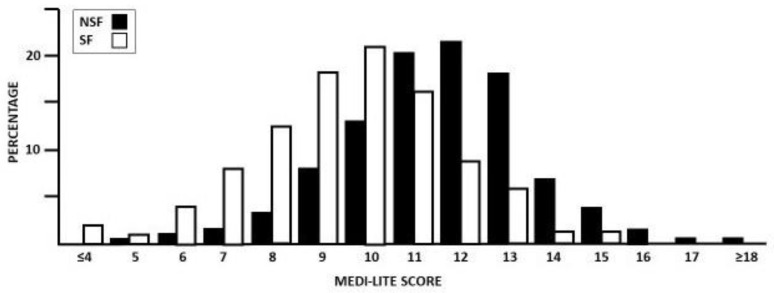
Distribution of stone-forming patients (SF) and non-stone-forming subjects (NSF) in percentages, according to MED-LITE score values.

**Figure 2 nutrients-16-00783-f002:**
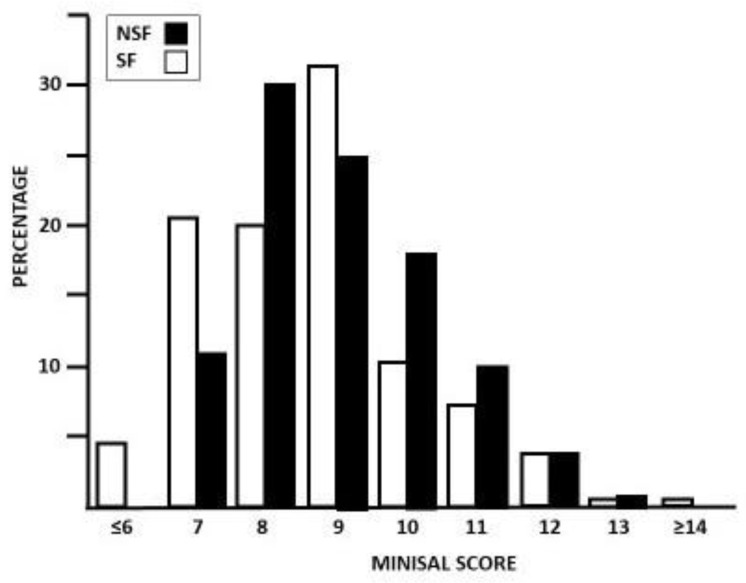
Distribution of stone-forming patients (SF) and non-stone-forming subjects (NSF) in percentage, according to MINISAL score values.

**Table 1 nutrients-16-00783-t001:** Clinical characteristics of the study population classified according to stone-forming and non-stone-forming patients.

	SF	NSF
Number (*n*)	255	535
Male/Female (*n*; %)	117; 45.9:138; 54.1	245; 45.8:290; 54.2
Age (years)	53.1 ± 10.4	55.1 ± 12.7
BMI (kg/m^2^)	27.7 ± 4.97	26.7 ± 4.98
Schooling (*n*; %)	167; 65.5	222; 41.7 ^a^

Data are expressed as mean ± standard deviation for continuous variables, and as absolute number; percentage for dichotomic variables. SF: stone-forming patients. NSF: non-stone-forming patients. Number: number of the subjects enrolled for each group. BMI: body mass index. Schooling: subjects who obtained an upper secondary school diploma. ^a^: Significantly different compared to SF.

**Table 2 nutrients-16-00783-t002:** Answers to MEDI-LITE questionnaire in study populations.

	Stone-Forming Patients (*n* = 255)			Non-Stone-Forming Patients (*n* = 535)		
Food Consumption	High	Medium	Low	High	Medium	Low
Fruit	49; 19.3	95; 37.2	111; 43.5	133; 24.9 ^a^	252; 47.1 ^a^	150; 28.0 ^a^
Vegetables	20; 70.8	144; 56.6	91; 35.6	82; 15.3 ^a^	324; 60.6 ^a^	129; 24.1 ^a^
Cereal	34; 13.3	156; 61.2	65; 25.5	27; 13.5 ^a^	116; 58.0 ^a^	57; 28.5 ^a^
Legumes	53; 20.8	166; 65.1	36; 14.1	251; 46.9 ^a^	225; 42.1 ^a^	59; 28.5 ^a^
Fish	29; 11.4	140; 54.9	86; 33.7	102; 19.1 ^a^	327; 61.1 ^a^	105; 19.6 ^a^
Meat	21; 8.3	110; 43.1	124; 48.6	37; 6.9 ^a^	137; 25.6 ^a^	361; 67.5 ^a^
Dairy Products	30; 11.8	107; 42.0	118; 46.2	23; 4.3 ^a^	123; 23.0 ^a^	389; 72.7 ^a^
Alcohol	8; 3.1	229; 89.8	18; 7.1	1; 0.2 ^a^	459; 85.8 ^a^	75; 14.0 ^a^
Olive Oil	200; 78.4	40; 15.7	15; 5.9	503; 90.0 ^a^	3; 5.6 ^a^	2; 0.4 ^a^

Data are expressed as absolute number; percentage. For the variables fruit, vegetables, cereals, legumes, and fish, two points were assigned in the case of high consumption, one point in case of medium, and zero in case of low. For the variables meat and dairy products, two points were assigned in the case of low consumption, one point in case of medium, and zero in case of high. For alcohol, two points were assigned in the case of medium consumption, one point in case of low, and zero in case of high. For olive oil, two points were assigned in the case of high consumption, one point in case of medium, and zero in case of low. ^a^: Statistically significant compared to SF.

**Table 3 nutrients-16-00783-t003:** Answers to MINISAL questionnaire in study populations.

	SF (*n* = 255)			NSF (*n* = 535)		
Salt Consumption	High	Medium	Low	High	Medium	Low
(1) How often do you use salt at table?	54; 21.2	49; 19.2	152; 59.6	54; 10.1 ^a^	133; 24.9 ^a^	348; 65.0 ^a^
(2) How much bread do you eat in one day?	82; 32.2	172; 67.4	1; 0.4	81; 15.1 ^a^	413; 77.2 ^a^	41; 7.7 ^a^
(3) How many times a week do you eat cheese and/or cold cuts?	15; 5.9	139; 54.5	101; 39.6	35; 6.5 ^a^	207; 38.7 ^a^	293; 54.8 ^a^
(4) Do you ever get thirsty especially after a meal?	26; 10.2	63; 24.7	166; 65.1	32; 6.0 ^a^	201; 37.6 ^a^	302; 56.4 ^a^
(5) When you eat out of home, does food seem usually salty, normal, or bland?	37; 14.7	173; 67.8	45; 17.6	136; 25.4 ^a^	362; 67.7 ^a^	37; 6.9 ^a^

Data are expressed as absolute number; percentage. SF: stone-forming patients. NSF: non-stone-forming patients. Salt consumption: estimated salt consumption. For variables from 1 to 4, three points were assigned in the case of high consumption, two points in case of medium, and one in case of low. For variable 5, three points were assigned in the case of a bland perception, two for normal, and one for salty. ^a^: Statistically significant compared to SF.

**Table 4 nutrients-16-00783-t004:** Study populations dichotomized by median values of MEDI-LITE (11 points) and MINISAL (9 points) scores.

Group	MEDI-LITE	MINISAL	SF (*n* = 255)	NSF (*n* = 535)	*p*
A	>11	<9	43; 16.9	211; 39.4	<0.05
B	>11	>9	45; 17.6	179; 33.5	
C	<11	<9	61; 23.9	61; 11.4	
D	<11	>9	106; 41.6	84; 15.7	

Data are expressed as absolute number; percentage. SF: stone-forming patients. NSF: non-stone-forming patients.

## Data Availability

Data is contained within the article.

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
