# Peer review of "Adherence to Mediterranean Diet, Dietary Salt Intake, and Susceptibility to Nephrolithiasis: A Case–Control Study"

_nutrients, 2024, doi:10.3390/nu16060783_

Round 1

Reviewer 1 Report

Comments and Suggestions for Authors

Abate et al. aim with this case study to assess i) adherence to the Mediterranean Diet (MD) and dietary salt intake in stone patients (SF), II) the relationship between adherence to the Mediterranean Diet (MD) and dietary salt intake in stone disease (CS) patients, II) the relationship between adherence to the MD salt intake and NL-related metabolic risk factors in FS, and iii) the impact of the combination of high salt intake on susceptibility to NL. I consider it an interesting research that can provide practical input, I congratulate the authors. I report below on possible improvements.

·       consider including a section on practical applications

·       consider improving the conclusions section by adding the most relevant of the above practical applications.

·       There are some references from the 2000s that could improve the document.

Author Response

Reviewer: Abate et al. aim with this case study to assess i) adherence to the Mediterranean Diet (MD) and dietary salt intake in stone patients (SF), II) the relationship between adherence to the Mediterranean Diet (MD) and dietary salt intake in stone disease (CS) patients, II) the relationship between adherence to the MD salt intake and NL-related metabolic risk factors in FS, and iii) the impact of the combination of high salt intake on susceptibility to NL. I consider it an interesting research that can provide practical input, I congratulate the authors. I report below on possible improvements.

Authors: We thank the reviewer for taking your time in reading our manuscript. Your tips are very intersting, and we believe that they could improve the quality of the manuscript. 

Reviewer:   consider including a section on practical applications

Authors: As suggested, a practical application section (section 5) has been added in the manuscript, and also the same have been discussed in the conclusion (section 6). 

Reviewer: consider improving the conclusions section by adding the most relevant of the above practical applications.

Authors: the conclusion section has been ameliorated as suggested (section 7)

Reviewer: There are some references from the 2000s that could improve the document.

Authors: The reference list has been updated. 

Reviewer 2 Report

Comments and Suggestions for Authors

The study uses the correct point method to assess the compliance of the examined people's diet with the Mediterranean diet. Also, the method of assessing the reliability of estimating salt intake was correct because it was confirmed by urinary sodium excretion within 24 hours. The authors also drew correct conclusions.
I am missing an assessment of urine pH in this work, because it is an important risk factor for the formation of kidney stones, especially in acidic urine. Urine pH is related to the composition of the diet.  This parameter should appear in subsequent works. You can add a few sentences to the introduction about the influence of dietary ingredients on urine pH.

Author Response

Reviewer: The study uses the correct point method to assess the compliance of the examined people's diet with the Mediterranean diet. Also, the method of assessing the reliability of estimating salt intake was correct because it was confirmed by urinary sodium excretion within 24 hours. The authors also drew correct conclusions.
I am missing an assessment of urine pH in this work, because it is an important risk factor for the formation of kidney stones, especially in acidic urine. Urine pH is related to the composition of the diet.  This parameter should appear in subsequent works. You can add a few sentences to the introduction about the influence of dietary ingredients on urine pH.

Authors: The sentences about the impact of acid load on diet and urinary stones has been added, both in the introduction (page 2, line 53-57) and in the discussione (page 8, line 28-29)

Reviewer 3 Report

Comments and Suggestions for Authors

The issue of diet and nephrolithiasis is clinically very important so the study addresses an important issue.

The main weakness of this manuscript is the lack of novelty.  Adherence to the Mediterranean diet and stone propensity has been previously published. A compelling argument for what this particular study adds to the previous manuscripts is needed.

In addition, the authors need to describe how this study fits in with the current state of investigation on this topic.

Comments on the Quality of English Language

Some of the word usage is incorrect or awkward but could be easily fixed.

Author Response

- Reviewer: The issue of diet and nephrolithiasis is clinically very important so the study addresses an important issue.

Authors: Dear reviewer, we appreciated your comment to our study. 

- Reviewer: The main weakness of this manuscript is the lack of novelty.  Adherence to the Mediterranean diet and stone propensity has been previously published. A compelling argument for what this particular study adds to the previous manuscripts is needed.

Authors: We thank you for the opportunity to discuss the novelty from our study. Indeed, it has been reported in the discussion section (page 8, line 268-272)

Reviewer: In addition, the authors need to describe how this study fits in with the current state of investigation on this topic.

Authors: Later to the discussion of the novelty, the setting in which the study fits in has been argued. (page 8, line 271-272)

Round 2

Reviewer 1 Report

Comments and Suggestions for Authors

The authors have added what was requested so I consider that the article can be published, congratulations for the improvements. 

Author Response

Thank you!

Reviewer 3 Report

Comments and Suggestions for Authors

One final clarification would be critical.

Is the Mediterranean diet considered a low salt diet when followed appropriately?

If not, does the nutrition community have a sense of the average amount of salt consumed on the Mediterranean diet under excellent adherence compared to the DASH diet or other low salt diets?

If the Mediterranean diet IS considered a low salt diet, then can the authors actually state that the individuals ingesting high salt were adherent to the diet?

Author Response

Dear reviewer, a sentence regarding the contraddiction between the MD dietary pattern and salt intake of our patients has been added in the discussion section (page 8, line 286-289).